# Development of Digital Teaching Competence: Pilot Experience and Validation through Expert Judgment

**Juan Jesús Gutiérrez-Castillo** [1] , **Antonio Palacios-Rodríguez** [1,*] , **Lorena Martín-Párraga** [1] and **Manuel Serrano-Hidalgo** [2]

1   Facultad de Ciencias de la Educación, Departamento de Didáctica y Organizacion Educativa, Universidad de Sevilla, 41004 Sevilla, Spain
2   Secretariado de Innovación Educativa, Universidad de Sevilla, 41004 Sevilla, Spain
*   Correspondence: aprodriguez@us.es

**Abstract:** The development of digital teaching competence is one of the fundamental requirements of what is known as the "knowledge society". With the aim of evaluating, from an expert point of view, the design of a training itinerary oriented to the training development of non-university teachers under the t-MOOC architecture for the improvement of digital teaching competence (DTC), the following quantitative research is presented. For this purpose, a quantitative validation design was established using the expert judgement technique. To measure the level of the expert coefficient, the expert competence (K) index was calculated for a random sample of 292 subjects participating in the study: teachers belonging to preschool and primary schools in the Andalusian community. The responses of those experts who scored $\geq 0.8$ on the external competence index were then selected. The results demonstrate the validity of the tool produced (T-MOOC) as well as the uniformity of the criteria of the experts participating in the evaluation. Consequently, the necessary structuring of personalised training plans supported by reference models is discussed.

**Keywords:** digital competence; DigCompEdu; teacher training; MOOC; expert judgement; media evaluation; learning with ICT

## 1. Introduction

Today's society is immersed in profound and diverse changes all brought about by advances in the use of information and communication, thus generating a new technological era.

This technological context generates new ways of communication and, therefore, the emergence of leadership styles and frameworks that govern the expansion of new cultures, which are progressively challenging [1].

This impact generated in society creates collaborative and communicative environments, which also generate improvements in the educational field [2,3]. Despite this, mere inclusion in this technological era does not equalize the opportunities for access and use for the user, causing visible inequalities in the different levels of competence [4]. These demands require constant updates, generating uncertainties largely due to the lack of knowledge of what the immediate future will be like.

The strong presence of ICT in the field of education is progressively increasing, meaning that the digital competences that teachers must possess go beyond the mere mastery of teaching content and methodologies, as indicated in the Horizon Reports [5].

This call led the European Commission in 2006 to consider digital competence as essential for the critical and safe use of ICT. It is understood as a set of transversal competences that a society must possess in order to be able to function effectively in the knowledge society [6].

In response to the demands raised, in 2013, the European Union Commission detailed the importance of "reorienting education" as a way to achieve teaching excellence within

current environments, which led to the introduction of ICT in the educational framework. This led to the creation of international training plans capable of effectively integrating the CD among teachers, ensuring common training in this competence [7]. It could be argued, therefore, that ICT has managed to acquire a relevant role as an indispensable resource among teachers, whose level of competence will be essential in providing quality education in virtual environments [8].

The introduction of these means and resources by teachers requires constant reflection on the aspects that may hinder their correct application. According to [9], it is necessary to review the concept of literacy and to advance in new forms of identification that facilitate greater and better access to the development of the competences demanded by society. The incessant need to achieve the introduction of a literacy model generates digital culture promoting digital literacy, e-learning, e-inclusion, e-health, and digital solutions in these fields.

Due to the existing demands and the need to develop digital competences and strategies, official bodies and institutions began to draw up a list of essential competences, among which we find digital competence [10] and with it new expressions, such as digital teaching competence (DTC). This definition refers to the development of knowledge, skills, and strategies to address educational problems using digital technology [11,12].

This teaching in digital competences crosses several dimensions and therefore needs to offer reference frameworks for its correct application, as detailed by [13]. Among them we find the ISTE standards, the UNESCO framework, and the INTEF framework—each and every one of them were analysed in different studies.

In our field of study, we will focus, as detailed above, on the one provided by the European Union: the DigCompEdu framework (Digital Competences Framework for Educators). This framework has a total of six areas (Figure 1).

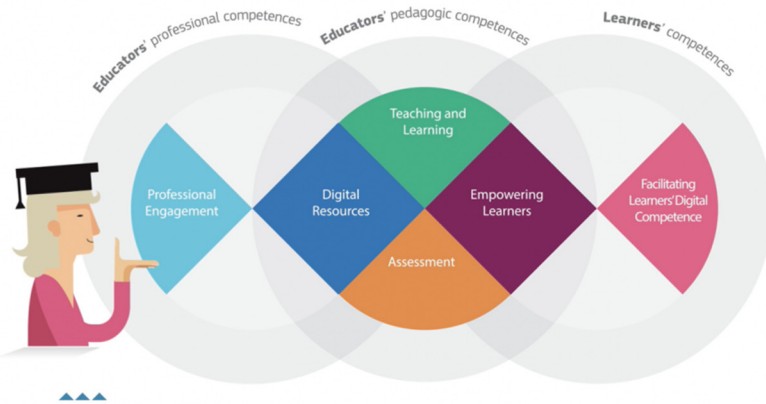

**Figure 1.** DigCompEdu Framework. JRC.

DigCompEdu is a competency model of six areas (Figure 2) that are associated with different competences that teachers must acquire in order to achieve the promotion of productive, inclusive, and integrative learning strategies through the use of digital tools [14].

As proposed by [15–20], they can (a) offer global learning where learner participation and engagement would come into play; (b) enable access to public and shared content, thus generating more emergent knowledge; (c) have an impact on higher educational stages; and (d) improve educational quality and design. MOOCs could therefore be said to represent an impetus to enhance and promote the 2030 Agenda and the SDGs [21].

As we have previously pointed out, there are currently many reference frameworks around digital competence that have emerged in recent years. These frameworks must be understood as a guide to orient the development of said competence. They must go further and serve as a basis for the development of learning strategies that induce and/or favour the digital development of individuals.

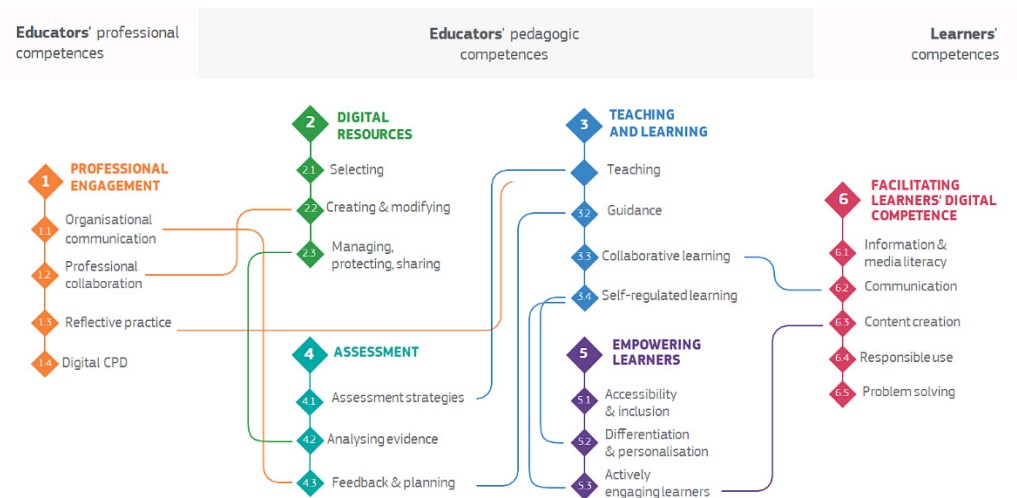

**Figure 2.** Overview of the DigCompEdu Framework. JRC.

Using one of the most representative frameworks currently available regarding digital competence, such as DigCompEdu, this research is presented. The research questions that arise are as follows: Can training itineraries be designed to promote digital competence based on the DigCompEdu framework? Are t-MOOCs configured as valid and reliable training instruments for the development of said competence?

## 2. Methods

### 2.1. Objective

The main objective of this research was to carry out an expert evaluation of a training design for the development of digital teacher competence (DTC) using the DigCompEdu reference framework.

### 2.2. Context/Material

An evaluation of a t-MOOC training environment developed for the further development of DTC following the DigCompEdu framework was offered. The platform where the content was developed was Moodle (https://www.dipromooc.es/ accessed on 1 December 2022). Among the advantages of this LMS, we can highlight [22]:

- Diverse possibilities for developing greater independence, commitment, and exercise of the learner;
- The use of various digital resources that improve skills in the use of information;
- The possibility of generating new knowledge, exchanging learning experiences, and searching for information.

The evaluation of the training action generated for the improvement of DTC under the DigCompEdu framework is presented below. Each of the competence areas included a video pill describing them. After viewing the video, the educator started with an elaboration of the content and ended with the completion of a final activity or task. A total of 4 to 6 activities oriented to each competence and level were available, from which the teacher had to choose to carry out 2 of them.

A didactic guide was provided showing how to identify the tasks, make recommendations for use, and measure the excellence of the distribution (Likert 1–6 points) via a checklist, and a final rubric was used by tutors in the final evaluation.

T-MOOC included tasks of different kinds: creation of concept maps, participation in group forums, design of blogs, construction of personal learning environments (PLE), elaboration of e-activities for students, construction of learning communities, among others.

We could say that t-MOOC included [23]:

1. A total of sixty-six learning modules (3 per competence);

2. Two hundred and thirty e-activities included in the modules;
3. One general didactic animation;
4. Six specific didactic animations per competence area;
5. Twenty-two specific animations for each competence;
6. Twenty-four infographics for the different modules;
7. Eleven polymedia for the different learning modules.

### 2.3. Procedure

The sample selected was random, and the questionnaire was sent to all ICT coordinators in publicly funded preschools and primary schools of the Department of Educational Development and Vocational Training of the Andalusian Regional Government (Spain).

A total of 364 e-mails were sent for data collection. The instrument was made available to teachers for two weeks, and a total of 292 responses were obtained. A total of 52% of the subjects surveyed were male compared to 48% who were female; the average age was 50 years.

In order to measure the level of expert knowledge [24,25] of the sample members, the experts were asked to self-assess their own level of knowledge [26] based on criteria such as publications on the subject to be analysed, years of experience, training, professional career, etc.

In this sense, the expert competence index (K) used in studies of a similar nature [27,28] was calculated. To calculate this index, the following parameter was used: K = $\frac{1}{2}$ (Kc + Ka), where (Kc) is the "Knowledge Coefficient" that the expert has about the research topic and (Ka) is the "Argumentation Coefficient" or the sources of criteria of each expert. The result of the CCE index yielded a value ≥0.8, which is considered a high value of expert competence [26].

In this case, to refine the expert selection process, we selected those experts who scored ≥ 0.8. This allowed us to identify 50 experts; this represented 17.12% of the total number of responses obtained. The proportion of the selected sample based on the gender variable is very similar to the proportion of responses obtained at a general level.

Fifty-nine percent of the experts were male compared to 41% who were female. The average age of the experts was 54 years. Although all of them work in preschools and primary schools, 24% (12 subjects) of them also work as associate professors in Andalusian public universities.

Finally, the vast majority of the experts selected (92.4%) indicated that they had teaching experience with, had published on, or had participated in research or working groups related to ICT and/or digital literacy and competence, both for teachers and learners.

### 2.4. Instrument

To meet the objectives of our study, we adapted the instrument designed in [29] to assess the self-concept, management, and planning of other technology. We selected this instrument based on adequate internal consistency, both at a general level and in each of its individual dimensions, corroborated by other studies that have shown the importance of these dimensions in the university population.

The designed instrument consisted of a total of 18 items distributed in four dimensions (see Table 1) related to the architecture of the designed t-MOOC.

Cronbach's Alpha statistic was applied to calculate the reliability with a very acceptable result (0.985) for the instrument as a whole. With regard to reliability by dimensions, the data also show high levels of reliability: technical and aesthetic aspects (0.996), ease of use (0.901), diversity of resources and activities (0.956), and quality of content (0.979).

The instrument used a Likert-type scale with six response options: 1. MN = Very negative/Very negative/Strongly disagree/Very difficult; 2. N = Negative/Disagree/Difficult; 3. R− = Fairly negative/Moderately disagree/Moderately difficult; 4. R+ = Fairly positive/Moderately agree/Moderately easy. The items are grouped into 4 dimensions and are shown in Table 1.

The questionnaire was administered online using the Google Forms tool: https://cutt.ly/PzZsfCV (accessed on 11 October 2022).

**Table 1.** Dimensions and items of the instrument.

| **1. Technical and aesthetic aspects** |
| --- |
| 1.1. The functioning of the MOOC we have presented to you is:<br>1.2. Overall, the aesthetics of the MOOC produced you would rate it as:<br>1.3. Overall, the technical functioning of the MOOC produced would you rate it as:<br>1.4. Overall, how would you rate the presentation of the information on the screen: |
| **2. Ease of use** |
| 2.1. How would you rate the ease of use and handling of the MOOC we have presented to you?<br>2.2. How would you rate the ease of understanding of the technical operation of the MOOC we have presented to you:<br>2.3. From your point of view, how would you rate the overall design of the MOOC we have produced:<br>2.4. From your point of view, how would you rate the accessibility/usability of the MOOC we have presented to you:<br>2.5. From your point of view, how would you rate the flexibility of use of the MOOC we have presented to you: |
| **3. Diversity of resources and activities** |
| 3.1. The diversity of resources used in the MOOC facilitates the understanding of the contents:<br>3.2. The materials, readings, animations, videos . . . offered in the MOOC are clear and appropriate:<br>3.3. The structure and materials of the MOOC are motivating to study:<br>3.4. The activities offered in the MOOC are attractive and innovative:<br>3.5. There are different modalities and types of activities: reinforcement, support and extension activities presented in the MOOC. |
| **4. The quality of the content** |
| 4.1. The contents of the MOOC and the structure are clear and appropriate.<br>4.2. The contents presented in the MOOC are appropriate to the competences to be acquired<br>4.3. The level of difficulty of the MOOC contents is easy to understand. |

## 3. Results

Before presenting the results, the reliability of the questionnaire was checked by applying Cronbach's Alpha (0.925) and McDonald's Omega (0.963) statistics. Both showed excellent results in terms of the overall reliability of the expert questionnaire.

The mean standards and their corresponding standard deviations achieved within the four areas that make up the data tool in addition to an overall assessment are detailed below (Table 2).

**Table 2.** Mean rating data and standard deviation obtained by experts.

| Areas | M | SD |
| --- | --- | --- |
| Technical aspects | 5.23 | 0.659 |
| Simplicity of use | 5.17 | 0.725 |
| Variety of resources and activities | 5.19 | 0.763 |
| Quality of content | 5.41 | 0.669 |
| Total | 5.25 | 0.630 |

The data obtained corroborate that in each of the evaluations of each dimension, the dimension has been judged positively. In particular, the high quality of the contents (5.41) and the technical aspects of the t-MOOC (5.23) stand out. The rest of the dimensions also maintained good scores in terms of simplicity of use and the variety of resources and activities contained in the t-MOOC.

Below, we can see the scores obtained for the different items in each of the dimensions.

With regard to Dimension 1 (technical and aesthetic aspects), two fundamental, interrelated aspects should be noted (see Table 3): It is worth highlighting the high score obtained by the experts both in terms of the functioning of the t-MOOC itself and its technical functioning, considering that the tool's functioning was correct and adequate and allowing users to navigate through its architecture easily and comfortably.

**Table 3.** Mean score and standard deviation of the experts for the items of the dimension "Technical and aesthetic aspects".

| Technical and Aesthetic Aspects | M | SD |
|---|---|---|
| The functioning of the t-MOOC presented to you is as follows: | 5.37 | 0.715 |
| Overall, you consider the aesthetics of the t-MOOC produced: | 5.03 | 0.978 |
| Overall, the technical performance of the t-MOOC produced would you rate it as: | 5.39 | 0.746 |
| Overall, how would you rate the presentation of the information on the screen? | 5.13 | 0.873 |

Table 4 shows the data resulting from the mean and standard deviation of the second dimension referring to the ease of use:

**Table 4.** Mean score and standard deviation of the experts for the items of the dimension "Technical and aesthetic aspects".

| Ease of Use | M | SD |
|---|---|---|
| How would you rate the ease of use and handling of the t-MOOC we have presented to you? | 5.34 | 0.799 |
| How would you rate the ease of understanding of the technical operation of the t-MOOC we have presented to you? | 5.32 | 0.822 |
| From your point of view, how would you rate the overall design of the t-MOOC we have produced? | 5.14 | 0.868 |
| From your point of view, how would you rate the accessibility/usability of the t-MOOC we have presented to you? | 5.27 | 0.841 |
| From your point of view, how would you rate the flexibility of use of the t-MOOC we have presented to you? | 5.26 | 0.823 |
| Using the t-MOOC produced was fun for you? | 4.70 | 1.165 |

It should be noted that most of the scores obtained in the items were high, where M >5, with the exception of the item "Using the t-MOOC produced was fun", which obtained a lower score (4.70). This fact shows the need for a review of the tool's architecture to make it more attractive to future users.

Along the same lines as the scores obtained in the previous dimensions, the values obtained in Dimension 3 (diversity of resources and activities) (see Table 5) exceeded the average score of 5. The high score achieved in the item "The materials, readings, animations, videos . . . offered in the t-MOOC are clear and appropriate" (5.29) may lead to the conclusion that the materials selected are suitable for producing quality learning situations. Similarly, the score achieved in the item "From your point of view, how would you evaluate the accessibility/usability of the t-MOOC we have presented to you?" (5.27) may lead to the conclusion that the architecture designed for the t-MOOC favours accessibility and/or usability for users, an aspect of vital importance in the design of this type of materials.

The mean values of the scores of the last dimension of the instrument (the quality of the contents) are presented in Table 6. It should be noted that the scores obtained in each of the items of this dimension reached the highest scores in the instrument.

**Table 5.** Mean score and standard deviation of the experts on the items of the dimension "Diversity of resources and activities".

| Diversity of Resources and Activities | M | SD |
|---|---|---|
| The diversity of resources used in the t-MOOC facilitates the understanding of the contents. | 5.17 | 0.927 |
| The materials, readings, animations, videos . . . offered in the t-MOOC are clear and appropriate. | 5.29 | 0.917 |
| The structure and materials of the t-MOOC are motivating for study. | 5.09 | 0.946 |
| The activities offered in the t-MOOC are attractive and innovative. | 5.17 | 0.920 |
| There are different modalities and types of activities: reinforcement, support, extension . . . presented in the t-MOOC. | 5.25 | 0.775 |

**Table 6.** Mean score and standard deviation made by the experts on the items of the dimension "Quality of contents".

| The Quality of the Content | M | SD |
|---|---|---|
| The contents of the t-MOOC as well as its structure are clear and adequate. | 5.40 | 0.810 |
| The contents presented in the t-MOOC are appropriate to the competences to be developed. | 5.44 | 0.752 |
| The contents of the t-MOOC are easy to understand. | 5.40 | 0.700 |

## 4. Discussion

The importance of this research is supported by the effectiveness of the procedure used. The evaluations carried out by the experts allowed for considerable improvements in some of its aspects. In this sense, to highlight those crucial aspects in the architecture of the t-MOOC that have obtained very high scores, such as navigation and the technical functioning of the t-MOOC, the improved version of the training action will include:

- Less linear structure.
- Modification of some tasks.
- Presentation of contents including complementary material.

Authors such as [30] point out that MOOCs should have a clear course structure where navigation and orientation is facilitated.

## 5. Conclusions

Different researchers support ways of designing training actions by considering the use of dissimilar tools used for the exposition of data and carrying out activities in each of the modules to overcome and reach the next level [31]. This way of developing training actions makes it necessary to rethink new ways of designing the exact resources used in online training [32–34]. It should be noted that this tool, according to the assessments offered by the experts, offers the possibility of training teachers in DTC within the DigCompEdu framework, and therefore supports the proposed training plan.

Therefore, the acquisition of the skills demanded by the digital society in which we find ourselves is favoured [35,36].

Finally, the pilot experience can increase and lead institutions regarding the strategies to follow in order to establish strategies for teacher training in digital competences for teachers. In this sense, the present study offers the possibility of looking at the training environment as a training plan that, thanks to its architecture, is offered for teacher training through a training offer of different levels linked to the European Framework for Digital Competence in Education: DigCompEdu [37].

The main limitations of this study are related to the use of self-perception-based questionnaires. However, the large number of responses obtained as well as the expert selection process through the use of the expert competence coefficient increased its scientific potential. However, as a line of future research, the use of qualitative evaluation instruments is recommended. Some lines of consideration could be: qualitative evaluation by the experts

or qualitative evaluation by the students participating in the training action. Likewise, the evaluation of this environment has been carried out almost exclusively by experts of Spanish nationality. Therefore, it is suggested to repeat the experience with international experts. In this case, a prior contextualization should be produced in the training action.

**Author Contributions:** Conceptualization, J.J.G.-C. and A.P.-R.; methodology, A.P.-R.; software, M.S.-H.; validation, L.M.-P., J.J.G.-C. and M.S.-H.; formal analysis, A.P.-R.; investigation, J.J.G.-C.; writing—review and editing, L.M.-P.; funding acquisition, J.J.G.-C. All authors have read and agreed to the published version of the manuscript.

**Funding:** This research was funded by Ministry of Science, Innovation and Universities (Spain), grant number RTI2018-097214-B-C31.

**Institutional Review Board Statement:** Not applicable.

**Informed Consent Statement:** Informed consent was obtained from all subjects involved in the study.

**Data Availability Statement:** The data is on the project page https://grupotecnologiaeducativa.es/ (accessed on 1 December 2022).

**Conflicts of Interest:** The authors declare no conflict of interest.

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
