# Peer review of "Development of Digital Teaching Competence: Pilot Experience and Validation through Expert Judgment"

_education, doi:10.3390/educsci13010052_

Round 1
Reviewer 1 Report
This paper reports the evaluation of a MOOC about digital competences. The paper has a thoroughly designed methodology, which is clearly explained, and provides credible results.
The paper is very interesting and can make a reasonable contribution, but it has some weaknesses that should be addressed:
- The paper needs a much more clearly stated research problem/research aims/research questions, and they need to be better aligned with the discussion and conclusion. This one of the reasons for a "reconsider after major revision" recommendation.
- The discussion and conclusion are very limited, both in scope and length. The statements proposed there need much more elaboration. Also, the limitations and future work of the study must be clearly stated. This the main reason for a "reconsider after major revision" recommendation.
- The sentences across the whole article are too long and with too many commas, which makes it a bit difficult to read at times.
- In the abstract, reduce the intensity of the first sentence, which can come across as an overstatement
- Digital Competence in Teaching (CDD): The initials are in Spanish. Translate the acronym too, or at least state the language of the intials.
- In 2.2. Context/Material. Do not start with "therefore", it does not make sense.
- You obtained 292 responses, 50 of which were deemed as experts. You need to state the gender distribution of the overall respondents as well and determine whether there is a significant difference between the gender ration of the whole sample and the gender ratio of the 50 selected. Also, clarify a bit more that the expertise level is self-reported, especially if the aforementioned difference is significant. There is a lot of literature about the gender gap in self-promotion, and it may be worth considering it.
- "The designed instrument consisted of a total of 18 items distributed in four dimen- 148 sions (see Table 1) related to the architecture of the designed t-Mooc. Reliability. 149 Croncbach's Alpha statistic was applied, with a very acceptable result (.985) for the instru- 150 ment as a whole" This passage is very disjointed, and there are many more of those. You need to proofread more thoroughly.
Overall, this is an article with quite a strong potential, but it seems as if it had been produced too quickly. It could certainly benefit from a bit more work in most of its aspects.
Reviewer 2 Report
The article is coherent, although it needs a review of its writing. The topic is very interesting and covers a relevant topic. At first glance, it seems to offer a significant perspective on the study of teacher digital competence, with a great contribution to the educational community. However, it has some aspects that could be improved.
It is recommended:
Restructure the introduction. The theoretical foundations studied should be deepened. It is recommended to add new references that highlight the research to date on the subject (teaching digital competence) and models (INTEF, DigCompEdu, ISTE...).
Round 2
Reviewer 1 Report
Comments in review addressed satisfactorily